# Chirality transfer from a 3D macro shape to the molecular level by controlling asymmetric secondary flows

Semih Sevim[1,2,10], Alessandro Sorrenti [1,3,4,10✉], João Pedro Vale [5,6,10], Zoubir El-Hachemi[3], Salvador Pané [2], Andreas D. Flouris [7], Tiago Sotto Mayor [5,6✉] & Josep Puigmartí-Luis[4,8,9✉]

Homochirality is a fundamental feature of living systems, and its origin is still an unsolved mystery. Previous investigations showed that external physical forces can bias a spontaneous symmetry breaking process towards deterministic enantioselection. But can the macroscopic shape of a reactor play a role in chiral symmetry breaking processes? Here we show an example of chirality transfer from the chiral shape of a 3D helical channel to the chirality of supramolecular aggregates, with the handedness of the helical channel dictating the direction of enantioselection in the assembly of an achiral molecule. By combining numerical simulations of fluid flow and mass transport with experimental data, we demonstrated that the chiral information is transferred top-down thanks to the interplay between the hydrodynamics of asymmetric secondary flows and the precise spatiotemporal control of reagent concentration fronts. This result shows the possibility of controlling enantioselectively molecular processes at the nanometer scale by modulating the geometry and the operating conditions of fluidic reactors.

[1] Institute of Chemical and Bioengineering, Department of Chemistry and Applied Biosciences, ETH Zurich, 8093 Zurich, Switzerland. [2] Multi-Scale Robotics Lab, ETH Zurich, Tannenstrasse 3, CH-8092 Zurich, Switzerland. [3] Departament de Química Inorgànica i Orgànica (Secció de Química Orgànica), University of Barcelona (UB), 08028 Barcelona, Spain. [4] Institut de Química Teòrica i Computacional, University of Barcelona (UB), 08028 Barcelona, Spain. [5] Transport Phenomena Research Centre (CEFT), Engineering Faculty of Porto University, Rua Dr. Roberto Frias, 4200-465 Porto, Portugal. [6] Associate Laboratory in Chemical Engineering (ALICE), Engineering Faculty of Porto University, Rua Dr. Roberto Frias, 4200-465 Porto, Portugal. [7] FAME Laboratory, Department of Exercise Science, University of Thessaly, Volos, Greece. [8] Departament de Ciència dels Materials i Química Física, University of Barcelona (UB), 08028 Barcelona, Spain. [9] Institució Catalana de Recerca i Estudis Avançats (ICREA), Pg. Lluís Companys 23, 08010 Barcelona, Spain. [10]These authors contributed equally: Semih Sevim, Alessandro Sorrenti, João Pedro Vale. ✉email: asorrenti@ub.edu; tiago.sottomayor@fe.up.pt; josep.puigmarti@ub.edu

Homochirality is a fundamental feature of animate matter. It has a key role in determining the structure and function of macromolecules and supramolecular assemblies in biological systems. Proteins, for example, are solely composed by L-amino acids while nucleic acids (DNA and RNAs) are comprised of D-sugars. Yet, the origin of homochirality in these functional structures remains a major mystery in the scientific community[1].

Recently, the observation that an enantiomeric excess (e.e.) can spontaneously emerge during the self-assembly of an achiral molecule into a chiral aggregate—or during its crystallization into a chiral space group—due to a chiral symmetry breaking process has fuelled extensive research in the supramolecular chemistry area[2–8]. In particular, it has been shown that external physical forces such as circularly polarized light[9–11] and/or macro- and microscopic mechanical forces[6,12–16] can bias a spontaneous symmetry breaking process towards a deterministic enantioselection. Note that if no bias is applied, a fully stochastic sign of the generated e.e. is accomplished[4,8,17]. However, in all these studies, the mixing of reagents and the nucleation/aggregation events are not controlled in time and space, and hence, they are not triggered in a rational manner. That is, the role of mass transport and concentration gradients in chiral selection phenomena, and in the top-down transfer of chiral information, has been essentially overlooked in the literature. Yet, the spatial and temporal control over the reagents' mixing via controlled diffusion is known to have a strong effect on the outcome of self-assembly processes[18–23], and particularly on chirality induction phenomena[24,25].

Herein, we show that the emergence of enantioselectivity during the assembly of an achiral molecule does not simply rely on a single external force, or a combination of forces, as recently reported[12,13], but rather on a series of physical and chemical constrains that act synergistically, and in a step-wise fashion, across multiple length scales. By combining numerical simulations of fluid flow and mass transport with a series of validation experiments, we demonstrate that we can rationally control a chiral symmetry breaking process occurring in a helical device. This is achieved by manipulating the secondary flows generated inside the device so as to localize the reaction zone (RZ) and the nucleation events in one of the two counter-rotating vortices that characterize the secondary flow.

The flow inside helical devices is more complex than in straight devices because the curved nature of the walls induce centrifugal forces that are not entirely balanced by the pressure gradients[26], thus resulting in secondary flows in the plane perpendicular to the main flow, that are characterized by one or two counter-rotating vortices[26,27]. The dimension, orientation and intensity of these counter-rotating vortices depend on the Reynolds number of the flow and on the curvature and torsion of the helical devices (see below). This in turn affects the mass transport and the concentration patterns developing along the helical devices. Interestingly, because the asymmetry between the upper and lower vortices in the secondary flow is known to increase with increasing torsion of the helix[26,28,29] and decreasing Reynolds number of the flow[30], we can leverage these effects to bias a symmetry breaking process in the assembly of an achiral molecule, by ensuring that the RZ for the aggregation is exposed primarily to one of the two vortices.

## Results and discussion

**3D helical devices.** 3D helical channels with a cross-section diameter of 1 mm were fabricated employing a commercially available 3D-printer (see Methods for further details). Two geometries were considered to induce different levels of asymmetry in the flow, one with a short pitch ($p = 1.5$ mm; left device in Fig. 1a) and another with a long pitch ($p = 7.5$ mm; right device in Fig. 1a), for the same curvature radius of the helix ($r = 1.5$ mm; Supplementary Fig. 1). Both geometries were prepared in their right-handed (R) and left-handed (L) enantiomeric forms, resulting in four devices $R_{short}$, $R_{long}$, $L_{short}$, $L_{long}$, each long enough to ensure four helical turns (i.e., the angle $\theta$ in Fig. 1b is related to the longitudinal position along the helix, and is, at most, $4 \times 360°$). In addition, for comparison, a device comprising a linear channel with the same diameter, hereafter called Linear, was also fabricated. 3D illustrations and geometrical parameters of all the used devices are reported in Supplementary Fig. 2, and Supplementary Table 1. The inlet region of the fluidic devices was designed to generate a concentric flow focusing, where a concentric sheath, created by the solution injected through two lateral inlets, surrounds the stream that is fed through the central inlet (Fig. 1c).

**Fluid flow and mass transport within the helical devices.** To investigate the flow field and the evolution of the concentration fronts generated within the $R_{short}$ and $R_{long}$ devices, we conducted numerical simulations of the flow and mass transport processes inside the devices. We considered flow rates implying laminar flows with low Reynolds number (Re ~ 30) to maximize the effect of pitch, and thus of torsion (Supplementary Fig. 1), over the flow asymmetry[30]. Moreover, we modelled the diffusive mass transport of a prototypical achiral porphyrin, i.e., 5-phenyl-10, 15, 20-tris(4-sulfonatophenyl)porphyrin (TPPS₃), which is frequently used as model system to study chiral symmetry breaking phenomena (Fig. 1d and Methods for further details)[12,13]. Mimicking the flow conditions considered in the experiments, the numerical simulations allowed us to track the position of the RZ as it moves along the $R_{short}$ and $R_{long}$ devices, and assess its exposure to the counter-rotating vortices (i.e., the secondary flow) forming inside the two helices (vide infra). Interestingly, the two helical devices studied induce vastly different flow patterns (Fig. 1e). The long-pitch device ($R_{long}$) leads to a strongly asymmetric secondary flow where a slightly rotated lower counter-clockwise (CCW) vortex occupies a much larger portion of the channel cross-section than the upper clockwise (CW) vortex (Fig. 1e, right). In sharp contrast, the secondary flow forming in the short-pitch device ($R_{short}$) reveals itself as a pair of counter-rotating vortices, which are almost symmetric relative to the plane passing through the center of the channel cross-section (Fig. 1e, left). Note that the fluid elements move along the helices in three-dimensional trajectories (Supplementary Fig. 3), and that the lines shown in Fig. 1e are just the two-dimensional (2D) representations of the above 3D trajectories, after a change of base (Supplementary Figs. 1, 3) to show the motion of the fluid elements relative to the centre of each cross-section (see Methods for further details). Note also that, as expected, the numerically simulated flow within the enantiomeric $L_{short}$ and $L_{long}$ devices showed flow patterns that mirror those obtained with the corresponding $R_{short}$ and $R_{long}$ devices, thus with the CW and cCCW vortices occupying opposite positions relative to the $R_{short}$ and $R_{long}$ counterparts (not shown).

More importantly, because of the very different flow patterns forming in the $R_{short}$ and $R_{long}$ helical devices, and the concentric flow focussing at the inlet, the injected porphyrin is exposed to dramatically different flow conditions in both helices (Fig. 1e). While in the $R_{short}$ device the injected porphyrin is exposed to both the CW and CCW vortices (Fig. 1e, left), in the $R_{long}$ device the injected porphyrin is exposed exclusively to the lower CCW vortex (Fig. 1e, right). This difference is of paramount importance, since it is at the basis of the dramatically different

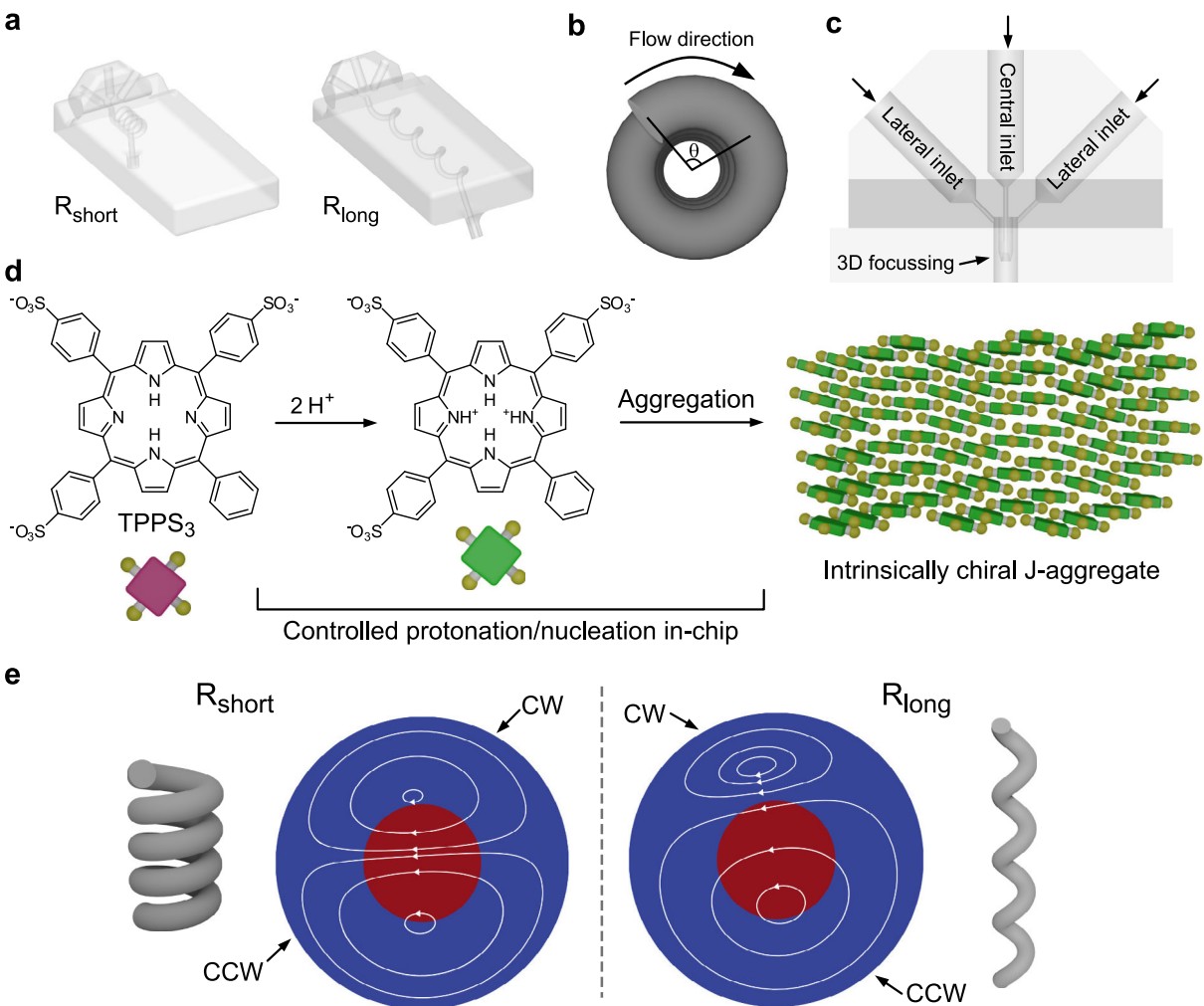

**Fig. 1 Representation of the fluidic devices and supramolecular system used and calculated fluid trajectories. a** Illustrations of $R_{short}$ and $R_{long}$ devices. **b** Definition of the rotational angle θ. **c** Detail of the inlet portion of fluidic devices. To achieve a concentric hydrodynamic 3D focussing, a 2 mm long concentric needle was designed to deliver the solution injected through the central inlet at the centre of the main channel, while surrounded by the sheath generated by the solutions fed through the lateral inlets. **d** Schematic of the protonation-induced aggregation of $TPPS_3$ to give intrinsically chiral J-aggregates (note: in the cartoon of the porphyrin the yellow balls represent the phenyl rings either bearing or not the sulfonate group). **e** 2D-representation of trajectories of the secondary flow forming inside the $R_{short}$ (left) and $R_{long}$ (right) devices, showing different relative magnitudes of the counter-rotating CW and CCW vortices in short- and long-pitch helices. The porphyrin is injected in the middle of the channel (red colour) surrounded by a sheath of the acid solution (blue region). Boundary conditions: 15 μM $TPPS_3$ at the central inlet, 10 mM HCl at the lateral inlets.

mass transport by advection that occurs along the two helical devices. Plotting the concentration maps of $TPPS_3$ (Fig. 2) at selected cross-sections along the entire length of the helical channel (i.e., four helix turns or 1440°) demonstrates that the concentration distribution evolves in a strikingly different manner in the $R_{short}$ and $R_{long}$ devices, as the porphyrin is transported downstream by the flow (see animation in Supplementary Movie 1). Although the porphyrin is injected at the middle of the channel and surrounded by a concentric sheath of the HCl/NaCl solution (θ = 0° in Fig. 2) in both devices, in the case of the $R_{short}$ device, the $TPPS_3$ stream splits between the upper and lower portions of the channel cross-section starting from θ = 180° (Fig. 2) because of the CW and CCW vortices prevailing in those regions, respectively, and remains similarly distributed between the two vortices for the remaining length of the helical channel (i.e., for 180° < θ < 1440°, Fig. 2). In stark contrast, the numerical simulations for the $R_{long}$ device show that the porphyrin remains 'trapped' in the CCW vortex, along the

entire helix (Fig. 2 and Supplementary Movie 1). Clearly, the different concentration distribution of $TPPS_3$ in both devices is the result of the markedly different hydrodynamic conditions induced by the secondary flows generated within the short-and long-pitch helices. Furthermore, note that although molecular diffusion plays a role in the mass transport inside the devices, causing the broadening of the porphyrin regions as they move along the channel (e.g., see the concentration maps at θ = 720° and θ = 1440° in Fig. 2), the very short residence time of the reaction solution in the devices (t < 2 s), joined with the low diffusion coefficient of the porphyrin, imply that molecular diffusion is not fast enough to spread $TPPS_3$ across the entire channel cross-section. Therefore, whether the porphyrin is exposed to one or both vortices as it moves along the helixes is strongly dependent on the asymmetry of the secondary flows induced by the helical devices. This, in turn, has a dramatic impact on the outcome of the chiral symmetry breaking observed in the aggregation process (*vide infra*).

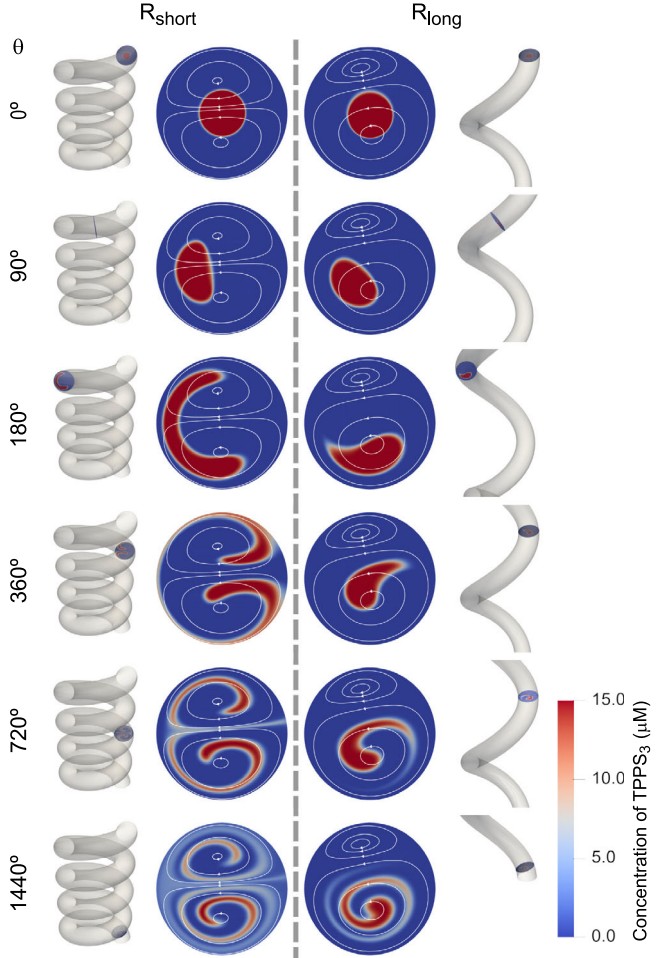

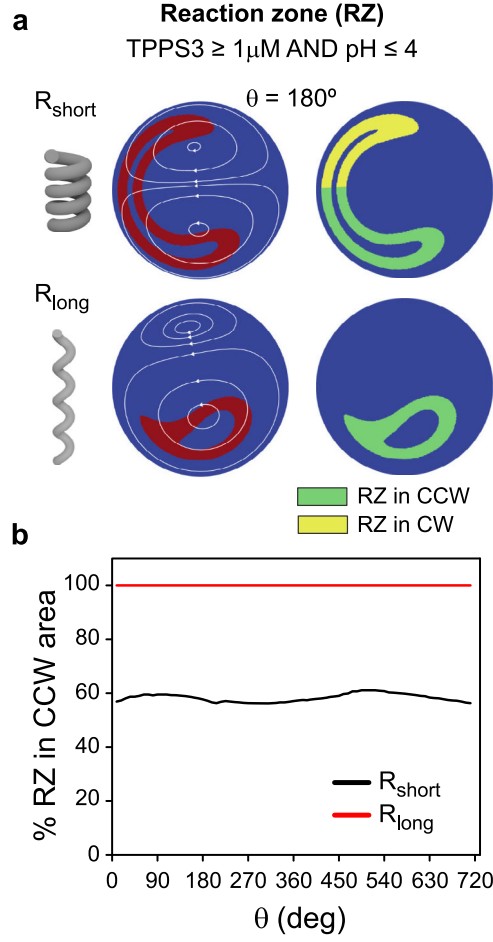

**Fig. 2 Mass transport in short- and long-pitch devices.** Concentration distribution of TPPS$_3$ at selected cross-sections along the entire length of the helical channel (four turns) for R$_{short}$ and R$_{long}$ devices (left and right, respectively). The first picture at $\theta = 0$ corresponds to the inlet region and shows the concentric 3D focussing. Snapshots taken from Supplementary Movie 1. The concentration distribution of TPPS$_3$ evolves in a markedly different way in the two devices, with the porphyrin remaining exposed exclusively to the CCW vortex in R$_{long}$ and to both the CCW and the CW vortices in R$_{short}$.

**Fig. 3 Evolution of the reaction zone along the helical channels. a** Definition of the reaction zone (RZ) and representation of the RZ calculated at the cross-section corresponding to $\theta = 180°$ for R$_{short}$ and R$_{long}$ devices (red colour in left panels). In the right panels the RZ is sorted in subregions depending on whether it is located in the CCW or CW vortex area (green and yellow colours, respectively). **b** Evolution of the fraction of the RZ exposed to the CCW vortex while the fluid moves downstream along the helix, in the R$_{short}$ and R$_{long}$ devices (two helical turns are reported for clarity, but analogous results were obtained for four turns).

**Positioning of the reaction zone (RZ).** The distribution of porphyrin concentration, per se, does not tell us much about the self-assembly process. The formation of porphyrin J-aggregates is a nucleated supramolecular polymerization driven by electro-static, hydrogen bonding and hydrophobic interactions, which is triggered in aqueous acidic media by the protonation of the porphyrin ring (and by the increase of ionic strength)[12,13,31]. To get further information on the position where porphyrin proto-nation can occur with respect to the two regions of opposite chirality, that is where J-aggregates can start to nucleate, we determined a reaction zone (RZ) defined as the region where porphyrin concentration is at least 1 μM and pH ≤4. We assume that under these conditions TPPS$_3$ porphyrin will be protonated to a great extent since its pKa is 4.8[32], and that the concentration of the protonated species would be sufficient for nucleation to happen. Figure 3a shows the RZ at the cross-section corre-sponding to $\theta = 180°$, for the R$_{short}$ and R$_{long}$ devices. Additionally, we sorted the RZ in two subregions depending on whether it lays within the CCW or the CW vortices (i.e., green and yellow colours in Fig. 3a, respectively; further details in the Methods section). The results show that, while in the R$_{long}$ device the whole RZ remains within the CCW vortex region along the entire length of the helical channel, in the R$_{short}$ device, roughly 60% of the RZ is located within the CCW region, with the rest being in the CW region; a distribution which remains mostly constant along the entire helix (Fig. 3b).

The constant distribution stems from the constant torsion and curvature of the helix, which produce flow patterns that are stable along its entire length. Furthermore, in laminar flows, fluid elements following a trajectory within one vortex should not move to a trajectory in another vortex. Therefore, because diffusion is not fast enough to significantly spread the RZ between the two vortices, a porphyrin molecule exposed to a given vortex should move along the entire helix under the influence of the same vortex. This is why the nucleation of J-aggregates is predicted to occur under the influence of both the CCW and the CW vortices (opposite chiral hydrodynamic fields) in the R$_{short}$ device, whereas it occurs under the influence of only the larger CCW vortex in the R$_{long}$ device (or of the CW vortex in the enantiomeric L$_{long}$ device).

In summary, the results from the numerical simulations show that one can precisely control the asymmetry of the secondary flows inside the helical devices, for instance by changing the pitch ($p$) of the helix while keeping constant the curvature radius ($r$) and Reynolds number of the flow (same flow rates and channel diameter), so as to ensure that the protonation of the porphyrin and the consequent nucleation events are located primarily in one of the counter-rotating vortices, as it occurs in the long-pitch devices considered in this work.

**From channel handedness to deterministic enantioselection.** To investigate experimentally the effect of the different secondary flow patterns (and RZ distribution) generated within the short- and long-pitch helical devices on the selection of supramolecular chirality, we studied the formation of J-aggregates of TPPS$_3$ triggered by mixing the porphyrin and the acid reactants inside the fluidic helical devices R$_{long}$, L$_{long}$, R$_{short}$, L$_{short}$, as well as within the straight device Linear. We anticipated that the experimental results will confirm the enantioselection that one can infer from the flow and mass transport predictions reported above.

J-aggregates of achiral TPPS$_3$ are intrinsically chiral species, where achiral monomers are arranged into a chiral crystalline lattice with a basic sheet geometry[6,33,34]. Spontaneous chiral symmetry breaking in those systems can lead to the stochastic formation of enantiomerically enriched mixtures of J-aggregates (scalemic mixtures), resulting in the emergence of natural optical activity. In the absence of any external chiral influence, the sign of the generated e.e. must be fully stochastic[4,8,17]. Conversely, deterministic enantioselection has been reported for J-aggregates of TPPS$_3$ in the presence of macroscopic vortices (i.e., where the sign of the detected e.e. correlates with the chirality of the vortex)[12,13]. A key requirement for this to happen is that the onset of self-assembly process occurs in the presence of chiral shear forces. These forces may act as a chiral polarization in a bifurcation phenomenon yielding a small e.e., which is then amplified during autocatalytic growth[3,14,33]. This scenario should not be confused with the reversible emergence of strong chiroptical signals occasionally observed when applying vortex stirring to fully-grown supramolecular assemblies, which can be ascribed to temporary alignment and/or sorting (e.g., by size and shape) of the aggregates in the hydrodynamic field[35–37]. Nor it should be confused with the mechanical resolution of chiral objects in chiral fluid flows[38,39].

In a typical experiment, we injected an aqueous solution of TPPS$_3$ (15 μM) into the central inlet of the fluidic devices, along with the injection of a solution of NaCl (125 mM) in aqueous HCl (10 mM, pH = 2) into the two lateral inlets. Typically, flow rates of 600/300/600 μL min$^{-1}$ were used, resulting in an effective residence time of the porphyrin in the fluidic device <2 s (Fig. 4a). The solution coming out from the device outlet was directly collected in a glass vial (3 mL) and the progress of the aggregation process was monitored by UV-Vis and circular dichroism (CD) spectroscopies over several days. The typical UV-Vis spectrum of the as prepared solutions (within 3 min) indicates the prevalent presence of monomeric diprotonated TPPS$_3$, as shown by intense absorption bands at 434 and 650 nm (Soret and first Q-band, respectively). In addition, a small band at 489 nm indicates the incipient formation of J-aggregates (Fig. 4a and Supplementary Fig. 4a). The intensity of the band at 489 nm strongly increases during a few hours after collection, alongside with the decrease of the intensity of the monomer's bands, indicating the fast aggregation of TPPS$_3$ in the collected solutions (Supplementary Fig. 4). On the other hand, CD bands emerge and evolve upon ageing of the J-aggregate solutions in the course of ~1 week

(Supplementary Fig. 5), without further changes of the absorption spectrum, in agreement with previously reported work on TPPS$_3$ aggregates undergoing chiral symmetry breaking[13]. Figure 4 summarizes the results obtained from the CD investigation. When comparing the enantiomeric long-pitch devices, R$_{long}$ and L$_{long}$, with the the Linear device, in all the cases, CD bisignate bands (exciton couplets) were observed in the region of the aggregate Soret band at 489 nm, which indicates the formation of a scalemic mixture of J-aggregates, where the sign of the CD couplets correlates with the sign of the enantiomeric excess (Fig. 4b, c)[40,41]. Nevertheless, enantioselection was clearly observed only in the J-aggregate solutions nucleated within the long-pitch helical channels, with the R$_{long}$ device promoting exclusively the appearance of positive (+/0/−) CD couplets, and the L$_{long}$ device leading to negative (−/0/+) CD couplets in more than 70% of the investigated samples (Fig. 4b, d). On the other hand, an almost statistical 50:50 distribution of positive and negative CD couplets was observed for the samples prepared using the Linear device, which is representative of a stochastic, unbiased, symmetry breaking (Fig. 4c, d). Furthermore, null CD spectra were occasionally observed when using the Linear device (black spectrum in Fig. 4c), indicating the sporadic formation of racemic mixtures of J-aggregates (with e.e. = 0). Finally, a closer look to the intensities of the CD spectra (Fig. 4d right) also reveals that the peak-to-peak amplitude of the positive and negative CD couplets observed in samples prepared, respectively, in R$_{long}$ and L$_{long}$ devices is higher than the amplitudes observed in the CD spectra of samples prepared in the Linear device, as well as higher than the amplitude of the positive CD couplets, observed sporadically when using L$_{long}$ devices. Taken together, the results indicate that when the nucleation of the J-aggregates occurs in the helical R$_{long}$ and L$_{long}$ devices, chiral sign selection takes place producing a deterministic bias in the supramolecular chirality of the aged TPPS$_3$ aggregates (as opposed to the stochastic emergence of chirality observed in Linear devices). Importantly, the direction of the bias is in agreement with the sense of the deterministic enantioselection previously reported for TPPS$_3$ aggregates in the presence of macroscopic vortices (with CCW and CW vortices inducing the appearance of positive and negative CD couplets, respectively)[12,13].

To summarize, in the case of the long-pitch devices (R$_{long}$ and L$_{long}$), we demonstrated that the information of chirality was ultimately transferred top-down from the 3D macroscopic shape (handedness) of the helical channels to the (supra)molecular level of the J-aggregates. This happened because the asymmetric secondary flows generated inside the devices, which depend on the interplay between the devices geometrical and operational parameters (curvature, torsion and Reynolds number), ensured the exposure of the RZ to only one of the two chiral vortices. In marked contrast, when J-aggregate solutions were prepared using the short-pitch helical devices (R$_{short}$ and L$_{short}$), similar three-signate CD bands were always recorded in all the samples investigated; thus, hampering the assignment of a specific supramolecular chirality (Fig. 4e). The shape of these CD bands is independent on the handedness of the used devices (Fig. 4e) and suggest the formation of "racemic" mixture of diastereomeric J-aggregates with almost opposite chiroptical features (i.e., superimposition of opposite CD couplets with slight different position)[42]. Note that the formation of different diastereomers (mesomorphs) with slightly displaced absorption bands has been previously reported for J-aggregates of trisulfonated porphyrins[33,43], which correspond to polymeric structures featuring different tacticity (i.e., orientations of the unsubstituted phenyl ring with respect to the sheet plane) and/or to incomplete sheet structures (oligomers)[33,34,43]. Here, it is plausible that distinct nucleation events take place in the

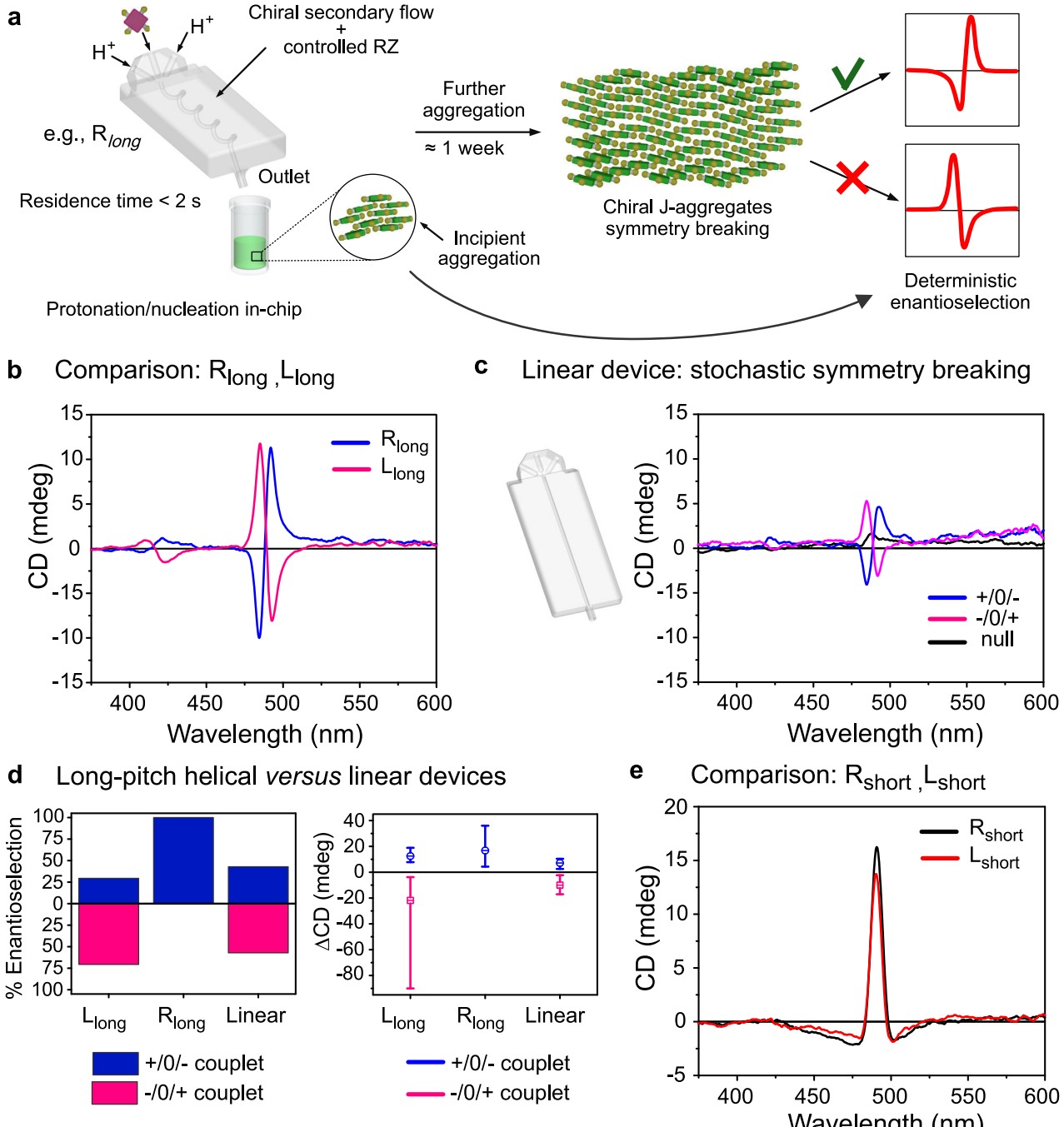

**Fig. 4 Enantioselection in TPPS₃ J-aggregate formation: overview of the experimental results. a** Schematic representation of the experimental procedure and of the deterministic enantioselection observed when using $R_{long}$ (or $L_{long}$) device. Mixing of the reactants, porphyrin protonation and aggregate nucleation occur during the short residence time in-chip (<2 s). Further aggregation and the appearance of CD bands occur overtime off-chip. **b** Representative CD spectra showing the opposite enantioselection obtained when using $R_{long}$ and $L_{long}$ devices. **c** Representative CD spectra obtained in the case of the experiments performed with the Linear device (stochastic symmetry breaking). **d** Left: stacked histogram representing the percentage of positive (blue) and negative (pink) CD couplets obtained when using $L_{long}$, $R_{long}$ and Linear microfluidic devices. Right: distribution of the peak-to-peak amplitudes for the positive (blue) and negative (pink) CD couplets observed in the same experiments. The centre of the squares indicates the average amplitude, the vertical bars represent the spreading of the amplitude values. **e** Representative CD spectra obtained in samples prepared with $R_{short}$ (black) and $L_{short}$ (red) devices showing trisignate bands.

subregions occupied by the CW and CCW vortices (that remain spatially separated along the entire helix), which may result into different growth pathways off-chip leading ultimately to slightly different mesomorphs[31,43].

In conclusion, we demonstrated an example of chirality transfer from the macroscopic chiral shape of a 3D helical channel to the chirality of supramolecular nanoassemblies, in which the handedness of the channel dictates the direction of enantioselection in the assembly of an achiral molecule. Numerical simulation of fluid flow and mass transport, combined with experimental data, allowed us to identify the conditions under which this top-down transfer of chiral

information can take place and to elucidate the mechanisms underlying it. The curved shape of a helical channel induces secondary flows in the form of two counter-rotating (CW and CCW) vortices, whose relative magnitude and orientation depend on geometric and operational parameters, such as the curvature and torsion (pitch) of the helix and the Reynolds number of the flow. By changing the pitch of the helix (e.g., while keeping constant the other parameters) we were able to control the asymmetry of the secondary flows in such a way that the RZ for the aggregation was exposed to only one of the two chiral vortices. This ultimately allowed us to bias the symmetry breaking process. While previous seminal reports demonstrated that chiral hydrodynamic fields can promote deterministic enantioselection in the formation of J-aggregates of the model porphyrin TPPS₃, the role of mass transport was completely overlooked (e.g., by premixing the reactants prior to the application of the chiral field)[12,13]. In contrast, by carefully controlling the reactant mixing as well as the RZ in space and time, we confirm that it is possible to transfer chiral information from an even higher level of chirality, that is, from the macroscopic handedness of the channel. In addition, we demonstrated that this transfer is mediated by the interplay of fluid mechanics and mass transport (advection and molecular diffusion) that act synergistically across the different scales. These findings generate the possibility of controlling enantio-selectively molecular process at the nanometre scale (including self-assembly and catalysis) via modulation of the geometry and operating conditions of fluidic reactors, provided that the transport phenomena (fluid flow and mass transport) are fully characterized.

## Methods

**Materials**. The trisodium salt of 5-phenyl-10,15,20-tris(4-sulfonatophenyl)por-phyrin, TPPS₃, was prepared according to a previously reported procedure[44]. NaCl for analysis was purchased from Sigma–Aldrich. HCl 37% was purchased from VWR International AG.

**Fabrication of microfluidic devices**. All devices were designed using a 3D CAD software (SolidWorks 2015). The manufacturing of microfluidic devices was per-formed with a 3D-printer (ProJet MJP 2500 Plus, 3D Systems Inc., U.S.) using a transparent resin (VisiJet M2R-CL, 3D Systems Inc., U.S.) via multi-jet printing (MJP) which is an inject printing process using piezo printhead technology to deposit either photocurable plastic resin or casting wax material layer by layer. After fabrication, the removal of wax was performed with MJP EasyClean System (3D Systems Inc., U.S.) followed by pressurized steam with a steam gun to elim-inate all the casting wax inside the microchannels.

**Microfluidic experiments**. All the inlet ports in the 3D printed microfluidic devices were designed to press-fit the PTFE microfluidic tubing (1/16" OD, 1 mm ID, BGB Analytik AG, Switzerland) for achieving chip-to-world interface. The solutions were sent into chip using a low-pressure syringe pump Nemesys 290 N (CETONI GmbH, Germany). Prior to all experiments, channels in microfluidic devices were washed with Milli-Q (MQ) water at a TFR of 600 µl/min for 15 min using syringe pumps.

In a typical experiment, an aqueous solution of TPPS₃ (15 µM) was injected into the central inlet of the fluidic devices (at a flow rate of 300 µl/min), along with the injection of a solution of NaCl (125 mM) in aqueous HCl (10 mM, pH = 2) into the two lateral inlets (at a flow rate 600 µl/min each). The solution coming out from the device outlet was directly collected in a glass vial (3 mL) and the progress of the aggregation process was monitored by UV-Vis and CD spectroscopies over several days.

**Circular dichroism (CD) and Uv-Vis spectra**. To measure CD spectra at different times (typically at 4 h, 3 days and 6 days after collection), 2.5 mL of the J-aggregate solutions were gently transferred to 1 cm quartz cuvettes (Hellma) by a micro-pipette just before the measurements. After each measurement, the solutions were then transferred back to the original glass vial and stored in dark. This procedure was aimed at preventing an overlong contact of the solutions with the quartz cuvette, which may result in the deposition of the aggregated material onto the cuvette walls and, consequently, in possible CD artefacts[35,36]. CD spectra were

recorded on static solutions on a Jasco J-815 spectropolarimeter. UV-Vis absorp-tion spectra were recorded on an Agilent Cary 60 spectrophotometer.

**Numerical simulations of transport phenomena**. The flow and mass transport inside the $R_{short}$ and $R_{long}$ devices was simulated using computational fluid dynamics. Velocity, pressure and species concentration were calculated using the finite volume method by coupling the Navier-Stokes equation for an incompres-sible Newtonian fluid, the continuity equation, and the species transport equation which are given by:

$$\frac{\partial \vec{V}}{\partial t} + \vec{V}(\nabla \cdot \vec{V}) = -\frac{1}{\rho}\nabla P + v\nabla^2 \vec{V} \quad (1)$$

$$\frac{\partial \rho}{\partial t} + \nabla(\rho\vec{V}) = 0 \quad (2)$$

$$\frac{\partial(\rho Y_i)}{\partial t} + \nabla(\rho\vec{V}Y_i) = \rho D_i\nabla^2 Y_i \quad (3)$$

where $\vec{V}$ is the velocity vector, $\nabla$ is the nabla operator, $\rho$ is the fluid density, P is the pressure, $v$ is the kinematic viscosity, $\nabla^2$ is the Laplacian operator, $Y_i$ is the mass fraction of species i and $D_i$ is the diffusion coefficient of species i.

The boundary conditions used in simulations mimicked the conditions used in the experiments. A 15 µM solution of porphyrin was introduced at the central device at a flow rate of 300 µL min⁻¹, and 10 mM solution of HCl was introduced at each of the lateral inlets at a flow rate of 600 µL min⁻¹. Because both the porphyrin and HCl flows are aqueous, the fluid properties were assumed to be those of water (density = 1000 kg m⁻³, viscosity = 0.001 kg m⁻¹ s⁻¹). Diffusion coefficients for the proton[45] and the porphyrin[46] were set to $5 \times 10^{-9}$ m² s⁻¹ and $3 \times 10^{-10}$ m² s⁻¹, respectively, based on literature data. No-slip condition was assumed at the walls of the helical channels, and atmospheric pressure was assumed at the device outlet.

A steady-state, double precision, pressure-based solver was used to solve the equations, considering the SIMPLE algorithm for the velocity-pressure coupling and using second order upwind spatial discretization for momentum and mass transport. Convergence was assumed when residuals were less than $10^{-6}$, with stricter criteria producing similar concentration and velocity fields.

To correctly visualize the mass transport inside the helical channel we used an appropriate orthogonal helical coordinate system based on the centre of each cross-section (rather than on the global reference frame, see further details in Supplementary Fig. 1)[27]. In this context, the 2D-representation of the 3D fluid trajectories depicted in Fig. 1 were obtained by connecting the points representing the positions of the fluid elements in each cross-section along the helical channel, and by plotting the 'connected points' in a single cross-section (see Supplementary Fig. 3 for details). This 2D-representation of the 3D secondary flow shows the motion of the fluid elements relative to the centre of each cross-section, and inform on how the reagents introduced at the inlet of the helical channels are moved around the cross-sections, as the flow proceeds along the device[47].

To determine the fraction of reaction zone under the influence of the CW and CCW vortices, we first plotted a large number of trajectories to identify the 'separation' between the vortices, for both devices. Fluid elements above that region were considered under the influence of the CW vortex, and those below it were considered are under the influence of the CCW vortex (for a R device). We identified the reaction zone as the region where the concentration of TPPS₃ was >1 µM and pH was ≤4 (red region in Fig. 3a), and then used an image analysis approach to determine the size of the CW and CCW portions of the reaction zone for each cross-section, based on the number of pixels in the CW and CCW regions. The fraction of the reaction zone under the influence of each vortex (Fig. 3b) was calculated by dividing the number of pixels in the CW and CCW portions of the reaction zone (Fig. 3a)

We conducted mesh-independence tests to identify the optimal mesh for the simulations in this work. The 3D model of the $R_{short}$ device was meshed with structured elements (to minimize artificial diffusion) featuring a total of 7, 12 and 14 million nodes, with the resulting velocity and porphyrin concentration profiles at various positions along the helix being compared in Supplementary Fig. 6. The results indicate that a structured mesh with 12 million nodes produces results that do not significantly differ from results obtained with denser meshes, and therefore all simulations were conducted based on 12 million node meshes.

For assessing the validity of the present modelling and simulation approach, we simulated the conditions reported in two experimental and numerical studies involving flows inside curved microchannels. We compared the results obtained with our model with those reported in the two studies in Supplementary Fig. 7, showing pressure drops along different channels for various flow rates[48] and shifts in the position of maximum velocity in channels with various Dean numbers[49]. The agreement between the predictions with the present model and the data from literature shows that the modelling approach followed in this this work can be used to accurately predict the flow patterns developing in curved microchannels.

## Data availability
The data that support the results of this study are available from the corresponding authors on request.

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

## Acknowledgements
This work is supported by the European Research Council Starting Grant microCrysFact (ERC-2015-STG No. 677020), the Swiss National Science Foundation (project no. 200021_181988) and grant PID2020-116612RB-C33 funded by MCIN/AEI /10.13039/501100011033. T.S.M. and J.P.L. also acknowledge support from the EU, from the Horizon 2020 FETOPEN project SPRINT (No. 801464).

## Author contributions
J.P.-L. conceived the idea. S.S. designed and optimized the helical devices and performed data analysis. A.S. designed and performed the experiments and performed data analysis. J.P.V., A.D.F., and T.S.M. performed numerical simulations of fluid flow and mass transport. Z.E.-H. synthesized TPPS₃ porphyrin. S.S., A.S., T.S.M. and J.P.-L. wrote the paper. S.S., A.S. and J.P.V. created the figures. All the authors discussed the results. S.S., A.S. and J.P.V. contributed equally to this work. A.S. and J.P.-L. led and supervised the investigation.

## Competing interests
The authors declare no competing interests.
