## [Peer Review File · Nature Communications]

REVIEWER COMMENTS

Reviewer #1 (Remarks to the Author):

I've read with interest the manuscript by Sevim et al. It reports detailed simulations of the hydrodynamic flow and mass transport inside helicoidal microfluidic channels of different pitch, together with experiments showing its implication on the chirality of self-assembled porphyrin aggregates. The results show that it is possible to control the handedness of supramolecular assemblies by changing handedness of the helicoidal microfluidic channel, its pitch and the flow rates. Moreover, a clear evidence of the importance of the chiral bias on the initial nucleation phase is provided. Previous results from the group of Prof. Minghua Liu[ref. 15] and of some of the Authors[25] showed somehow similar results, but with a completely different geometry of the microchannel. Also, in the present manuscript a very detailed and convincing analysis of the secondary flows inside the microchannel has been performed. The impact of the present investigation is high in the field of supramolecular chemistry, enantioselective synthesis and in the fundamental research on the homochirality of life.

The experiments seem carefully planned and performed, and the modelling of the hydrodynamic flow and mass transport is sound. Methods are fairly well described.

On these bases, I recommend publication of this manuscript on Nature Communications, after addressing the following minor points:

- i) The issue of the disentangled growth of the extinction feature at 490 nm and the corresponding CD band is very intriguing. On p. 7, l.217-220, the Authors point out a very slow evolution of the CD spectra with time (in contrast with the quite rapid increase of the 490 nm J-band in the UV/Vis extinction spectra), before reaching the equilibrium value. It could be very interesting to add a figure (even in Supporting Information) showing the development of this spectroscopic feature.
- ii) On p. 8, l. 237-238, the larger values for the negative CD couplets with respect to the positive ones is outlined (Fig. 4d). Do the Authors have any explanation for this observation? Could it be somehow related to some adventitious chiral contaminant or a chiral bias?
- iii) Minor typos: Fig. 1 l. 304 and 310, please correct the acronym of TPPS3

Reviewer #2 (Remarks to the Author):

The authors study the self-assembly process of an achiral molecule into a chiral compound. This self-assembly process is "triggered" by chiral shear forces which break the chiral symmetry and thus "select" the chirality of the final compound. The chiral shear forces are created in a helical fluidic channel by secondary helical flow fields. Investigating two different geometries, the authors show

(experimentally and by numerical solutions) that the dimensions of the helical channel can be adjusted such that the reaction zone for the molecules is predominantly located within a secondary vortex flow with defined handedness, providing the symmetry-breaking shear forces required for the chiral assembly process.

As far as I understand from the Introduction and the literature the authors cite, it has been known before that a chiral shear flow would break the symmetry and initiate a self-assembly process of a chiral aggregate, despite the individual constituent molecules being achiral. Compared to this literature, the central new addition in the manuscript seems to be the mass transport through the fluidic channel, in order to have more control over the chiral symmetry breaking process (Introduction, second and third paragraph).

This is definitely an interesting study, but I do not think the conceptual advance presented in the manuscript warrants publication in Nature Communications.

This judgement is based on the following considerations:

* The advantage resulting from the mass transport through the fluidic channel is not explained in the manuscript. Why is it better to have the reaction zone moving in a helical flow field through a channel, rather than simply stirring it in a pre-defined direction (selecting the chirality) within a "container", to generate the shear forces initiating the chiral self-assembly process?

* The setup the authors use seems to be unnecessarily complicated: Why using a geometry for the fluidic channel that in principle allows for secondary flows of both chiralities, rather than building a channel that creates a flow field with unique chirality, as for instance in [Science 295, 647-651 (2002)] or [Soft Matter 9, 2525 (2013)]?

* I do not understand what the authors intend to say with the first sentence of the third paragraph in the Introduction:

"Herein, we show that the emergence of enantioselectivity during the assembly of an achiral molecule does not simply rely on a single external force ... but rather on a series of physical and chemical constraints that act synergistically, and in a step-wise fashion, across multiple length scales".

Do they mean applying a chiral shear force would not be sufficient to initiate the chiral self-assembly? And what are the "physical and chemical constraints that act synergistically"?

A similar sentence can be found in the Conclusions.

* The authors state that "we demonstrate for the first time that we can rationally control a chiral symmetry breaking process occurring in a helical device" (third paragraph in the Introduction). I would not quite agree with that statement. There is considerable theoretical and experimental literature on chiral symmetry breaking processes that manifest not as chiral self-assembly but as

sorting of chiral entities, where the symmetry breaking process is induced by mechanical forces, see [Soft Matter, 15, 4593-4608 (2019)] and references therein, in particular [Soft Matter 9, 2525 (2013)] for a helical device).

Please find here below our point-by-point detail answers to the Referees' comments. The original reviewer's comments are in *black italic* and our reply in *blue*.

Reviewer #1 (Remarks to the Author):

I've read with interest the manuscript by Sevim et al. It reports detailed simulations of the hydrodynamic flow and mass transport inside helicoidal microfluidic channels of different pitch, together with experiments showing its implication on the chirality of self-assembled porphyrin aggregates. The results show that it is possible to control the handedness of supramolecular assemblies by changing handedness of the helicoidal microfluidic channel, its pitch and the flow rates. Moreover, a clear evidence of the importance of the chiral bias on the initial nucleation phase is provided. Previous results from the group of Prof. Minghua Liu[ref. 15] and of some of the Authors[25] showed somehow similar results, but with a completely different geometry of the microchannel. Also, in the present manuscript a very detailed and convincing analysis of the secondary flows inside the microchannel has been performed. The impact of the present investigation is high in the field of supramolecular chemistry, enantioselective synthesis and in the fundamental research on the homochirality of life. The experiments seem carefully planned and performed, and the modelling of the hydrodynamic flow and mass transport is sound. Methods are fairly well described.

We acknowledge Reviewer #1 for his/her comments which clearly highlight the novelty of our work and its high impact in the fields of supramolecular chemistry, enantioselective synthesis and biological homochirality.

On these bases, I recommend publication of this manuscript on Nature Communications, after addressing the following minor points:

i) The issue of the disentangled growth of the extinction feature at 490 nm and the corresponding CD band is very intriguing. On p. 7, l.217-220, the Authors point out a very slow evolution of the CD spectra with time (in contrast with the quite rapid increase of the 490 nm J-band in the UV/Vis extinction spectra), before reaching the equilibrium value. It could be very interesting to add a figure (even in Supporting Information) showing the development of this spectroscopic feature.

We thank the reviewer for his/her interesting comment. Yes, indeed the bisignate CD bands emerge slowly upon ageing of the solutions, in agreement with previously reported work on TPPS₃ aggregates undergoing chiral symmetry breaking. Following the suggestion of the reviewer, we added Figure S5 in Supplementary Information showing time course CD spectra for representative samples prepared using R_{long} and L_{long} devices.

ii) On p. 8, l. 237-238, the larger values for the negative CD couplets with respect to the positive ones is outlined (Fig. 4d). Do the Authors have any explanation for this observation? Could it be somehow related to some adventitious chiral contaminant or a chiral bias?

*We thank the reviewer for pointing this out. Actually, the average values of the intensities of the negative and positive CD couplets obtained upon deterministic enantioselection by using R_{long} and L_{long} devices are not very different (even if it is true that we sporadically obtained very intense negative couplets in the latter case). On the other hand, regarding the intensity of the CD bands, we think that the most relevant information that emerges from Figure 4d is that the average amplitude of the positive and negative CD couplets observed for samples prepared with R_{long} and L_{long} devices (via a successful enantioselection) is higher than the amplitudes of the CD couplets observed for samples prepared in the *Linear**

device (stochastic symmetry breaking), as well as higher than the intensity of the positive CD couplets observed sporadically when using L_{long} devices (unsuccessful enantioselection).

Some of us previously studied in detail the effect of chiral contaminants as possible biases in the spontaneous symmetry breaking observed during the formation of J-aggregates of the tetrasulfonate porphyrin TPPS₄ in the absence of chiral fields (Chirality, 2009, 408–412). While the presence of ubiquitous chiral contaminants cannot be discarded a priori, it seems that their potential effect is negligible in this study. In fact, we observed a statistical distribution of positive and negative CD couplets in samples prepared in the *Linear* device (spontaneous symmetry breaking), and successful deterministic enantioselection when using R_{long} and L_{long} devices.

iii) *Minor typos: Fig. 1 I. 304 and 310, please correct the acronym of TPPS3*

We thank the reviewer for these corrections that have now been included in our submitted revised version of the manuscript.

Reviewer #2 (Remarks to the Author):

The authors study the self-assembly process of an achiral molecule into a chiral compound. This self-assembly process is "triggered" by chiral shear forces which break the chiral symmetry and thus "select" the chirality of the final compound. The chiral shear forces are created in a helical fluidic channel by secondary helical flow fields. Investigating two different geometries, the authors show (experimentally and by numerical solutions) that the dimensions of the helical channel can be adjusted such that the reaction zone for the molecules is predominantly located within a secondary vortex flow with defined handedness, providing the symmetry-breaking shear forces required for the chiral assembly process.

As far as I understand from the Introduction and the literature the authors cite, it has been known before that a chiral shear flow would break the symmetry and initiate a self-assembly process of a chiral aggregate, despite the individual constituent molecules being achiral. Compared to this literature, the central new addition in the manuscript seems to be the mass transport through the fluidic channel, in order to have more control over the chiral symmetry breaking process (Introduction, second and third paragraph).

This is definitely an interesting study, but I do not think the conceptual advance presented in the manuscript warrants publication in Nature Communications."

We thank the Reviewer #2 for considering our study of interest. However, we kindly disagree with the reviewer when he/she states that our contribution lacks novelty.

Chiral symmetry breaking occurs spontaneously driven by random fluctuations, and in the presence of nonlinear dynamics, yielding chiral assemblies from achiral molecules (with stochastic sign of the generated enantiomeric excess, e.e.). Therefore, it does not require "symmetry-breaking shear forces", neither is the self-assembly "triggered by chiral shear forces" as stated by the reviewer. For example, in our linear devices where there are no chiral forces the symmetry breaking process also occurs and it is stochastic. It is true that previous seminal contributions in the field have shown that external physical forces, including chiral hydrodynamic vortices, can bias a spontaneous symmetry-breaking process towards deterministic enantioselection. However, the previous studies overlooked the role of concentration gradients and mass transport in the top-down transfer of chiral information (e.g., by premixing the reactants before the application of the chiral field). In our submitted manuscript, we have demonstrated that **the understanding and manipulation of the mass transport phenomena to precisely control the hydrodynamic conditions and the location of the reaction zone (RZ) can open the way for controlling enantioselectively molecular processes at the nanometer scale by simply modulating the geometry and the operating conditions of fluidic reactors.** Specifically, the control over mass transport did not merely

provide us with an improved control over symmetry-breaking as mentioned by Reviewer #2, but instead allowed us to achieve an unprecedented top-down chirality transfer from the macroscopic handedness of a 3D helical channel (i.e., an unmatched high level of chirality) to the chirality of supramolecular nanoassemblies, in which the handedness of the channel dictates the sign of the enantioselection. Lastly, our work offers, for the first time, an important insight into the mechanisms underlying the transfer of chiral information across different length scales, demonstrating that this transfer is mediated by the interplay of: i) the hydrodynamics conditions prevailing in the reactor (i.e. the asymmetric secondary flows generated in the chiral channel); ii) the mass transport phenomena occurring along the reactor (i.e. advection and diffusion); and iii) the spatiotemporal control of the concentration gradients and the RZ positioning (where aggregation is made to occur at precise locations and under specific hydrodynamic conditions). The full elucidation of the transport phenomena (fluid flow and mass transport) is crucial if one wants to control enantioselection at the nanoscale by playing with macroscopic variables, such as channel geometry and flow rates. Accordingly, and in our humble opinion, the results presented in our submitted manuscript are unprecedented and a breakthrough in the field.

This judgement is based on the following considerations:

** The advantage resulting from the mass transport through the fluidic channel is not explained in the manuscript. Why is it better to have the reaction zone moving in a helical flow field through a channel, rather than simply stirring it in a pre-defined direction (selecting the chirality) within a "container", to generate the shear forces initiating the chiral self-assembly process?"*

One should note that biasing a spontaneous symmetry breaking process towards a deterministic enantioselection remains a challenging task in the field of supramolecular chemistry, and certainly is not as easy as simply "stirring within a container", as suggested by Reviewer #2. In fact, it is crucial that the chiral polarization acts at the nucleation stage of the self-assembly process to generate a (non-stochastic) chiral bias, which is then amplified during autocatalytic growth. The few seminal reports that achieved deterministic enantioselection in a symmetry breaking process relied on a tedious and serendipitous optimization of the experimental conditions (see *Science* **2001**, 292, 2063–2066; *Nat Commun* **2018**, 9, 2599) and/or on the application of a combination of physical forces, such as rotation and strong magnetic fields, up to 25 T (*Nature chemistry* **2012**, 4, 201–207).

Against this background studies, we have shown that by thoroughly studying the fluid flow and the mass transport phenomena we could rationally control the chirality transfer by precisely controlling, in space and in time, the location of the RZ with respect to the asymmetric secondary flows. This control over the RZ positioning ultimately allowed us to control the chirality of the supramolecular assemblies depending on the handedness of the helical channel, and based on specific and rationally controlled geometrical and operational parameters.

We have described thoroughly in the entire manuscript, and further in our previous answers, the crucial importance of understanding and controlling the transport phenomena in order to rationally control the chirality transfer process. Therefore, and as also stated by Reviewer #1, we believe that our approach provides new groundbreaking insights on the study of chirality transfer phenomena.

** The setup the authors use seems to be unnecessarily complicated: Why using a geometry for the fluidic channel that in principle allows for secondary flows of both chiralities, rather than building a channel that creates a flow field with unique chirality, as for instance in [*Science* 295, 647-651 (2002)] or [*Soft Matter* 9, 2525 (2013)]?*

This comment indicates that Reviewer #2 misunderstood the groundbreaking nature of our study. In fact, the microfluidic device reported by Whitesides *et al.* ('*Chaotic Mixer for Microchannels*' Science 2002, 295, 647-651) was thought and designed (based on a non-trivial geometry optimization) to promote chaotic mixing to counter the poor solution homogenization and incomplete reactant mixing affecting microfluidic devices operating at low Reynolds number (laminar regime). In marked contrast, in our devices, **we take advantage of the incomplete reactant mixing and the formation of defined concentration gradients to achieve fine spatiotemporal control over the RZ positioning**, which in turn, **was crucial for achieving chirality transfer** (depending on the helical channel's pitch) and to get fundamental insight on this process. This concept is very well represented in Figs. 2 and 3 of the manuscript, showing how helical reactors enable a fine spatiotemporal control of the RZ positioning.

Additionally, we must point out that our device is much easier to operate than the microfluidic devices suggested by Reviewer #2 from the point of view of both geometry optimization and fabrication. The fabrication of the devices suggested by the reviewer requires multiple steps of photolithography in a cleanroom (for making the oblique ridges on the channel's floor) and additional soft lithography for making the PDMS slab. Any change of the design requires additional sessions of photolithography. On the other hand, our devices (with 1 mm cross-section channels) were fabricated out of a transparent resin in a straightforward manner by 3D printing using a commercially available printer (operating automatically overnight). Changes in design were easily performed by altering the CAD design and re-printing the device, which allowed us for a fast and inexpensive prototyping. For this reason, we do not agree that our device, a simple 3D printed helical channel, is "unnecessarily complicated", but rather the opposite. Finally, and in sharp contrast to the work mentioned by Reviewer #2 which aimed at increasing mixing, our helical reactor was developed to enable a precise control over the transport processes inside it, in order to demonstrate that, by understanding and controlling the transport phenomena (fluid flow and mass transport) in the reactor, we can rationally control the chirality transfer from a 3D macroscopic shape to the molecular level.

** I do not understand what the authors intend to say with the first sentence of the third paragraph in the Introduction:*

"Herein, we show that the emergence of enantioselectivity during the assembly of an achiral molecule does not simply rely on a single external force ... but rather on a series of physical and chemical constraints that act synergistically, and in a step-wise fashion, across multiple length scales".

Do they mean applying a chiral shear force would not be sufficient to initiate the chiral self-assembly? And what are the "physical and chemical constraints that act synergistically"?

A similar sentence can be found in the Conclusions.

This comment clearly shows that Reviewer #2 is not familiar with the concept of biasing (deterministically) a spontaneous symmetry breaking process.

Yes, we really mean that "applying a chiral shear force would not be sufficient to" lead to a deterministic chiral self-assembly. The use of asymmetric secondary flows (chiral field) may not be sufficient to achieve enantioselection (a non-trivial achievement), as we explained thoroughly in the paper and in the above answers. Indeed, it is necessary that the chiral polarization acts at the very early stages of the self-assembly process (at nucleation) to generate a bias. Furthermore, the distribution of porphyrin concentration does not give us enough information about the self-assembly process, as it is crucial to determine where the nucleation can happen with respect to the chiral vortices generated in the helical device. We did the latter by considering a putative reaction zone (RZ) based on specific chemical constraints, i.e. [porphyrin] > 1 μ M and pH \leq 4, and by tracking its evolution (positioning and spreading) along the entire helical channel. Note that, under those chemical conditions, it is

reasonable to assume complete porphyrin protonation and subsequent J-aggregate nucleation (as we detailed in the paper).

For example, when considering the *long-pitch* devices (R_{long} and L_{long}), the successful chirality transfer from their handedness to deterministic enantioselection occurs throughout the interplay of the following physical and chemical constraints:

1) **Fluid dynamics:** the macroscopic shape of the helical channel (torsion/pitch, curvature and handedness) and Reynolds numbers (flow rate and channel diameter) generate asymmetric secondary flows (unequal counter-rotating vortices)

2) **Mass transport:** the asymmetric flow pattern affects the mass transport by advection that, together with molecular diffusion, determine the concentration profiles of reactants developing along the helical devices

3) **Chemical constraints** define the RZ positioning, i.e. where nucleation is bound to occur with respect to the regions of different chirality. It is crucial that the nucleation occurs in-chip, i.e. during the short residence time in the device (< 2s)

Remarkably, as a result of our understanding of the combined action of these physical and chemical constraints, we were able to localize the RZ (nucleation events) in only one of the two counter-rotating vortices by simply modulating the pitch of the helical channel, which in turn, allowed us to induce a deterministic enantioselection. Note that, for example, the *short-pitch* devices did not yield the desired results even though, and in words of Reviewer #2, they also “apply a chiral shear force”.

** The authors state that "we demonstrate for the first time that we can rationally control a chiral symmetry breaking process occurring in a helical device" (third paragraph in the Introduction). I would not quite agree with that statement. There is considerable theoretical and experimental literature on chiral symmetry breaking processes that manifest not as chiral self-assembly but as sorting of chiral entities, where the symmetry breaking process is induced by mechanical forces, see [Soft Matter, 15, 4593-4608 (2019)] and references therein, in particular [Soft Matter 9, 2525 (2013)] for a helical device).*

We believe that the results we are reporting in our submitted manuscript have nothing to do, and must not be confused, with the mechanical resolution of chiral (macro- or micro-) objects in chiral fluid flows cited by Reviewer #2. In fact, they represent completely different physical phenomena. By analogy, that would be like confusing enantioselective synthesis with resolution of racemic compounds by chiral HPLC. In addition, we point out that the device reported in Soft Matter 9, 2525 (2013) is not helical, as claimed by the reviewer, but planar and similar to the Whitesides' device where a helical flow can be generated.

The spatial sorting of an enantiomeric object (featuring an obvious chiral shape) under the action of a chiral hydrodynamic field is based on the different (diastereomeric) trajectories they experiment in the field. Note that in these systems the chiral field must act continuously during the entire separation process. In marked contrast, we refer to spontaneous symmetry breaking as the process in which achiral molecules self-assemble yielding *spontaneously* to chiral aggregates, with stochastic sign of the generated e.e.. We have demonstrated that, depending on the macroscopic handedness of a simple helical channel, we can bias the symmetry breaking towards the desired supramolecular chirality, thus achieving deterministic enantioselection. This depends on geometric, operational and chemical conditions that we have fully rationalized. In addition, we would like to point out that, in the system we studied the action of the chiral hydrodynamic field (chiral polarization) occurs exclusively at the nucleation stage of the self-assembly and within the short residence time in the chip (< 2s). Accordingly, and to the best of our knowledge, this top-down transfer of chiral information

from the handedness of the channel to the chirality of supramolecular nanoassemblies has never been demonstrated before.

To further clarify the concerns raised by the reviewer, we have now included the following paragraph in our revised manuscript, including new references:

“This scenario should not be confused with the reversible emergence of strong chiroptical signals occasionally observed when applying vortex stirring to fully-grown supramolecular assemblies, which can be ascribed to temporary alignment and/or sorting (e.g. by size and shape) of the aggregates in the hydrodynamic field.^{36–39} Nor it should be confused with the mechanical resolution of chiral objects in chiral fluid flows.^{40,41}”

REVIEWERS' COMMENTS

Reviewer #1 (Remarks to the Author):

I've read the revised version of the manuscript by Sevim et al. and I feel that the authors have correctly addressed my previous points. I confirm my earlier comments on the impact of this work and that it represents an advancement in understanding the mechanism of chirality transfer from an external bias to a growing supramolecular system. It could be relevant for future development in this field. I recommend publication of the manuscript in the present form.

Reviewer #2 (Remarks to the Author):

The authors are correct: I was not really familiar with the concept of biasing a self-assembly process at nucleation stage to break its chiral symmetry. After having read the detailed reply by the authors and again the manuscript I now can appreciate much more the achievements reported in the manuscript, in particular the relevance of the mass transport in the helical device to control the reaction zone. I can well imagine "that biasing a spontaneous symmetry breaking process towards a deterministic enantioselection remains a challenging task in the field of supramolecular chemistry" (as the authors say), and the comments of the other referee confirm that the authors report on a major progress towards solving this task. I now agree on this assessment and therefore support publication of the manuscript in Nature Communications.

A few concluding comments:

* I was thinking about the setup from [Science 295, 647-651 (2002)] or [Soft Matter 9, 2525 (2013)], because the flow field created in these devices has a defined chirality, so that the reaction zone would always be located within a secondary flow with definite helicity. Would something like that not be advantageous? I imagine, when making the device a little large it could also be 3D-printed in the same way as the authors fabricate their helical structure.

* The argument about the mixing in these micro-fluidic channels is moot. It is correct that in [Science 295, 647-651 (2002)] the device has been devised with the aim to promote mixing, but in [Soft Matter 9, 2525 (2013)] it has actually been used for separation! Both modes of operation work because the channels are extremely long with tens or hundreds of periodically repeating chiral elements in the device. In contrast, in the current manuscript the fluidic channels is very short with only 4 turns (Fig. 2). However, one can anticipate in Fig. 2 (see 1440deg) from the spreading of the concentration distribution that for longer channels the helical device the authors use would actually mix.

* I did not say that mechanical chiral resolution is a physical phenomenon similar to the symmetry breaking process towards the desired supramolecular chirality the authors demonstrate. However, it is also an example in which a chiral symmetry breaking process is controlled in a helical device (more

precisely: a device with a "chiral" structure of definite handedness that generates helical flow fields), otherwise chiral separation would not be possible. In that sense I think the authors do not "demonstrate for the first time that we can rationally control a chiral symmetry breaking process occurring in a helical device".

REVIEWERS' COMMENTS

Reviewer #1 (Remarks to the Author):

I've read the revised version of the manuscript by Sevim et al. and I feel that the authors have correctly addressed my previous points. I confirm my earlier comments on the impact of this work and that it represents an advancement in understanding the mechanism of chirality transfer from an external bias to a growing supramolecular system. It could be relevant for future development in this field. I recommend publication of the manuscript in the present form.

We acknowledge Reviewer #1 once more for considering our work of relevance in the field of chirality transfer as well as for pointing out that our work can lead to new future developments in the field.

Reviewer #2 (Remarks to the Author):

The authors are correct: I was not really familiar with the concept of biasing a self-assembly process at nucleation stage to break its chiral symmetry. After having read the detailed reply by the authors and again the manuscript I now can appreciate much more the achievements reported in the manuscript, in particular the relevance of the mass transport in the helical device to control the reaction zone. I can well imagine "that biasing a spontaneous symmetry breaking process towards a deterministic enantioselection remains a challenging task in the field of supramolecular chemistry" (as the authors say), and the comments of the other referee confirm that the authors report on a major progress towards solving this task. I now agree on this assessment and therefore support publication of the manuscript in Nature Communications.

We highly appreciate the openness and transparency of the reviewer, and we truthfully acknowledge him/her for recognizing that our results represent a major step forward in the challenging task of harnessing spontaneous symmetry breaking processes.

A few concluding comments:

* I was thinking about the setup from [Science 295, 647-651 (2002)] or [Soft Matter 9, 2525 (2013)], because the flow field created in these devices has a defined chirality, so that the reaction zone would always be located within a secondary flow with definite helicity. Would something like that not be advantageous? I imagine, when making the device a little large it could also be 3D-printed in the same way as the authors fabricate their helical structure.

We understand the point raised by the reviewer, but we would like to point out again that the kind of devices reported by Whitesides et al. required a non-trivial geometry optimization (angle, depth, and number of the ridges) to achieve a vortical flow that occupy most of the cross sectional-area of the channel. We do not believe that simply making the device "a little larger" would automatically guarantee to have the same hydrodynamic field, conversely it would require a thorough new optimization, since scaling factors have a marked effect on fluid phenomena. For example, scaling up such a device to dimensions similar to those of our helical devices (i.e., 1 mm channel diameter) so that it could be 3D-printed, would correspond to scaling up its cross-sectional area by 56-fold (1.7 orders of magnitude) which would cause a huge change of the surface-to-volume ratio of the channel (i.e., Whitesides' device has the following dimension $70 \times 200 \mu\text{m} \rightarrow$ cross-sectional area = 0.014 mm^2 ; our device has a cross-sectional area of 0.786 mm^2). Moreover, our helical devices offer the advantage that the asymmetry of the secondary flows can be controlled by varying a simple and common parameter such as the helical pitch (e.g., while keeping constant the curvature and the Reynolds number). Furthermore, in our design, we could benefit of concentric hydrodynamic 3D focussing that, while allowing to control the positioning of the reaction zone (RZ) relative to the regions of different chirality (interplay of advection and reactant diffusion),

it also creates a concentric sheath around the porphyrin stream, and thus prevents the contact of the RZ with the channel walls. This contact could induce heterogeneous nucleation (i.e., at the channel walls), possibly detrimental to the control of the symmetry breaking process, besides likely leading to channel clogging. Please note that 3D flow focussing is not possible in a planar 2D device as the one suggested by the reviewer.

Finally, one of the main objectives and achievements of this paper has been demonstrating that, by understanding and harnessing the mass transport phenomena, it is possible to achieve unprecedented top-down chirality transfer from the macroscopic chirality of a channel, to the chirality of supramolecular nanoassemblies, in which the handedness of the channel dictates the sign of the enantioselection. As such, we believe that using a 3D helical channel featuring a well-defined chiral shape, familiar geometric parameters such as pitch and curvature radius, as well as a clear handedness (M or P), is the most obvious choice.

* The argument about the mixing in these micro-fluidic channels is moot. It is correct that in [Science 295, 647-651 (2002)] the device has been devised with the aim to promote mixing, but in [Soft Matter 9, 2525 (2013)] it has actually been used for separation! Both modes of operation work because the channels are extremely long with tens or hundreds of periodically repeating chiral elements in the device. In contrast, in the current manuscript the fluidic channels is very short with only 4 turns (Fig. 2). However, one can anticipate in Fig. 2 (see 1440deg) from the spreading of the concentration distribution that for longer channels the helical device the authors use would actually mix.

We agree with the reviewer that in [Soft Matter 9, 2525 (2013)] a device analogous to that of Whitesides has been used with a "separation" purpose. Nevertheless, the "separation" the reviewer is referring to consists in a spatial sorting of enantiomeric microscopic objects (featuring obvious chiral shapes) under the action of the hydrodynamic field. This kind of separation (as we previously explained) is based on the diastereomeric trajectories these objects experiment in the field. This phenomenon is not the opposite of the "mixing" promoted in the Whitesides' devices; indeed, they are two different phenomena. In fact, when considering a chemical reaction, e.g., flowing a reactant A and B in the device of [Soft Matter 9, 2525 (2013)], the effect of the flow field would be that of enhancing their mixing through stretching and folding of the fluid elements, exactly as in the case of the Whitesides' devices.

Besides, we politely disagree with the argument of the channel length. Although we are not sure to which length the reviewer is referring to, when the absolute length of the channel is considered, our device is longer than that of Whitesides. In fact, in [Science 295, 647-651 (2002)] the high number of ridges on the floor were inserted to decrease the mixing length (enhance mixing), which allowed them to have complete mixing in about 1 cm (e.g. in the staggered herringbone version of the device). In any case, their devices were maximum 3 cm long, whereas the arc-length of our helical channel is about 4.8 cm.

As correctly pointed out by the reviewer, by increasing the length of our channel even more (in reality much more!) molecular diffusion would inevitably lead to complete mixing. However, the key point is that we were not aiming at complete mixing. On the contrary, **we took advantage of the incomplete reactant mixing** and the formation of defined concentration gradients, to gain fine spatiotemporal control over the RZ positioning, which in turn was crucial for achieving chirality transfer.

* I did not say that mechanical chiral resolution is a physical phenomenon similar to the symmetry breaking process towards the desired supramolecular chirality the authors demonstrate. However, it is also an example in which a chiral symmetry breaking process is controlled in a helical device (more precisely: a device with a "chiral" structure of definite handedness that generates helical flow fields), otherwise chiral separation would not be possible. In that sense I think the authors do not "demonstrate for the first time that we can rationally control a chiral symmetry breaking process occurring in a helical device".

Given the context of our investigation (well-detailed along the entire manuscript), when we talked about “chiral symmetry breaking”, we were clearly referring to the process in which achiral molecules self-assemble yielding spontaneously chiral aggregates (or chiral crystals), with stochastic sign of the generated enantiomeric excess (*ee*). Put into context, it is undeniable that we have demonstrated for the first time that we can control symmetry breaking in molecular self-assembly, in such a way that the handedness of the helical channel dictates the sign of the final *ee*, which was based on the understanding and rational control of fluid flow and mass transport phenomena.

Having said that, to avoid controversy, we have now removed the statement “for the first time” from the manuscript. So, the sentence now reads:

By combining numerical simulations of fluid flow and mass transport with a series of validation experiments, we demonstrate that we can rationally control a chiral symmetry breaking process occurring in a helical device.